# Evaluating the Physiologic Effects of Alfaxalone, Dexmedetomidine, and Midazolam Combinations in Common Blue-Tongued Skinks (*Tiliqua scincoides*)

**DOI:** 10.3390/ani14182636

**Published:** 2024-09-11

**Authors:** Haerin Rhim, Ashleigh M. Godke, M. Graciela Aguilar, Mark A. Mitchell

**Affiliations:** Department of Veterinary Clinical Sciences, School of Veterinary Medicine, Louisiana State University, Baton Rouge, LA 70803, USA; hrhim1@lsu.edu (H.R.); agodke3@lsu.edu (A.M.G.); magui32@lsu.edu (M.G.A.)

**Keywords:** anesthesia, blood gas, blue-tongue skink, intubation, lizard, pH, PCO_2_, reptile, sedation, *Tiliqua scincoides*

## Abstract

**Simple Summary:**

Common blue-tongued skinks (*Tiliqua scincoides*) are popular pets due to their docile temper. Because of their popularity, they are routinely presented to veterinarians for examinations or procedures; however, to date, there has been limited research evaluating sedation protocols for this species. This study aimed to test different sedation combinations in these skinks: alfaxalone alone, alfaxalone with midazolam, dexmedetomidine with midazolam, and a combination of alfaxalone, dexmedetomidine, and midazolam. All four combinations provided safe sedation, but there were different physiologic responses noted. According to our trials, the combinations of all three drugs or alfaxalone with midazolam are recommended for minor procedures.

**Abstract:**

Common blue-tongued skinks (*Tiliqua scincoides*) are popular pet reptiles; however, there has been limited research to investigate sedatives for this species. The purpose of this study was to measure the physiologic effects of four combinations of alfaxalone, dexmedetomidine, and midazolam for minor procedures such as intubation and blood collection. Eleven common blue-tongued skinks (*Tiliqua scincoides*) were used for this prospective, randomized cross-over study. The subcutaneous combinations were used as follows: 20 mg/kg alfaxalone (A); 10 mg/kg alfaxalone and 1 mg/kg midazolam (AM); 0.1 mg/kg dexmedetomidine and 1 mg/kg midazolam (DM); and 5 mg/kg alfaxalone, 0.05 mg/kg dexmedetomidine, and 0.5 mg/kg midazolam (ADM). Heart rate, respiratory rate, palpebral reflex, righting reflex, escape reflex, toe pinch withdrawal reflex, tongue flicking, and the possibility of intubation were recorded at baseline and every 5 min for 60 min. Venous blood gases were measured at baseline, full sedation, and recovery. Heart and respiratory rates decreased significantly in all groups, but the reductions were most prominent in DM and ADM. Analgesic effects, as measured by the toe pinch withdrawal reflex, were only observed in DM and ADM. Intubation was possible in all four protocols; however, it was not possible in two DM skinks. Based on these trials, ADM and AM are recommended for minor procedures in blue-tongue skinks.

## 1. Introduction

Common blue-tongued skinks (*Tiliqua scincoides*) are popular as pets and exhibit animals at zoological institutions because of their inquisitive behavior and ease of care. Because of the popularity of these lizards, veterinarians are being asked to provide medical and surgical care for these animals. However, to date, there has been a dearth of evidence-based research related to the medical and surgical management of these animals. A search of PubMed and Google Scholar using the key words “*Tiliqua*” or “blue-tongued skink” on 1 July 2024 revealed only twenty-one articles related to the medical and surgical care of these animals, with five being case reports and sixteen being clinically related research. Limited evidence-based research can make it challenging for veterinarians to identify “best practices” for managing these animals in the captive setting.

One of the challenges a veterinarian faces when working with any new species is identifying safe, consistent sedation and anesthetic protocols to manage different diagnostic and surgical procedures. From the previously noted literature search, six papers were case reports or evidence-based articles for non-anesthetic purposes that described sedation or anesthesia in common blue tongue skinks, while one was an evidence-based research paper that used a specific anesthetic regimen on a study population of skinks [1,2,3,4,5,6,7]. In these seven articles, the sedation and anesthetic protocols were used to perform different procedures, including computed tomography scans, endoscopic sexing, radiography, and induction of anesthesia. The sedatives/anesthetics and doses used in these articles included propofol (9 mg/kg intravenous [IV] and 5 mg/kg IV) [1,2], medetomidine (0.2 mg/kg) and ketamine (1.1 mg/kg intramuscular [IM]) [3], alfaxalone alone (9 mg/kg IV and 10 mg/kg IM) [4,6], alfaxalone (10 mg/kg) and midazolam (0.4 mg/kg IM) [5], and midazolam (0.7 mg/kg), dexmedetomidine (0.05 mg/kg), and ketamine (5 mg/kg IM) [7]. The evidence-based study only evaluated alfaxalone, and the dose used was sufficient to obtain a loss of righting reflex for 9 min [4]. Based on these limited findings, it is important to pursue additional evidence-based research to identify safe and reliable sedation protocols for this species.

Alfaxalone is an injectable anesthetic commonly used in captive reptiles. Based on a search of the same engines and dates using the keywords alfaxalone and reptile, 42 original research studies have described the use of alfaxalone in reptiles as a solitary sedative or in combination with other sedatives [4,8,9,10,11,12,13,14,15,16,17,18,19,20,21,22,23,24,25,26,27,28,29,30,31,32,33,34,35,36,37,38,39,40,41,42,43,44,45,46,47,48,49]. Alfaxalone is a neuroactive steroid that acts on GABAA receptors as a positive allosteric modulator; it can also act as a direct agonist when used in high doses [50]. Major benefits attributed to alfaxalone are that it has minimal inhibitory effects on the cardiovascular and respiratory systems compared to other commonly used injectable anesthetics, and it has a rapid onset of action [12,51]. However, large volumes are needed in reptiles compared to other sedatives, which makes subcutaneous (SC) injection preferred over IM injection.

Dexmedetomidine, an α_2_-adrenoceptor agonist, is also a widely used injectable anesthetic in veterinary medicine. Based on a review of the same methods noted previously using the keywords dexmedetomidine, medetomidine, and reptile, 46 original research studies described using dexmedetomidine (*n* = 24) or medetomidine (*n* = 23) in reptiles [10,17,26,28,31,32,33,34,35,43,52,53,54,55,56,57,58,59,60,61,62,63,64,65,66,67,68,69,70,71,72,73,74,75,76,77,78,79,80,81,82,83,84,85,86,87]. The α_2_-adrenoceptor agonists have been found to provide sedation, muscle relaxation, and visceral analgesia, while cardiovascular and respiratory depression are the most commonly described adverse effects. Ineffective sedation has been reported in brown anole lizards (*Anolis sagrei*), Japanese grass lizards (*Takydromus tachydromoides*), and Argentine tegus (*Salvator merianae*) when dexmedetomidine was given as a single agent [32,33,70]. However, it is more common to combine α_2_-adrenoceptor agonists with other sedatives/anesthetics to reduce the dosages of each drug and minimize any potential adverse effects. Another benefit of α_2_-adrenoceptor agonists is that they can be reversed once a procedure is completed.

Midazolam is a commonly used benzodiazepine in veterinary medicine, and it is used both alone or in combination with other sedatives and anesthetics. The benzodiazepines enhance the effects of GABA on GABAA receptors, resulting in sedation. The most common uses for midazolam in veterinary medicine are as a premedication before surgery, as a sedative, and for seizure control [88,89]. According to the same search criteria mentioned previously and using the keywords midazolam and reptile, 36 original research studies were found describing midazolam use in reptiles [17,26,28,32,34,35,64,66,70,72,77,80,82,83,85,86,90,91,92,93,94,95,96,97,98,99,100,101,102,103,104,105,106,107,108,109]. As a sole agent, midazolam has been found to provide both successful and limited sedative effects in reptiles [32,91,93,99]; however, it is more common to combine midazolam with other sedatives or anesthetics to achieve a synergistic effect [70]. This also has been proven in pigs and rats, where the combination of α_2_-adrenoceptor agonists and midazolam provided maximal pain control for a longer duration than those used alone [110,111]. A major benefit of using multiple drugs in combination is that the adverse effects of each drug can be minimized by reducing the dose of each drug [112]. Another benefit of midazolam is that it can be reversed with flumazenil.

Given the limited data available for blue-tongue skinks, further evidence-based research is needed to better ascertain the value of sedatives for this species. The purpose of this study was to determine the preferred combinations and doses of alfaxalone, dexmedetomidine, and midazolam for providing safe and reliable sedation for performing basic procedures in blue-tongue skinks. The specific hypotheses for this study were that (1) all sedation protocols would be safe and provide sufficient sedation for minor procedures such as tracheal intubation, collecting radiographic images, and blood collection from the skinks; (2) sedation would occur within 10 min after SC injection; and (3) that the combinations with dexmedetomidine would result in decreased heart rates (HR) and respiratory rates (RR).

## 2. Materials and Methods

A prospective, crossover study was conducted under the regulations set forth by the Louisiana State University Institutional Animal Care and Use Committee (23-077).

### 2.1. Animals

Eleven adult (yearling) non-sexed common blue-tongued skinks were used in the study. The skinks were obtained from a private breeder and were used to routine handling. Physical examinations were performed on the skinks prior to the study, and the animals were found to be clinically healthy with mean ± standard deviation (SD) bodyweights of 666.8 ± 123.9 g. The animals were housed in 18” × 12” × 36” (45 × 31 × 91 cm) enclosures (Rubbermaid, Wooster, OH, USA) in a room with an ambient temperature and humidity of 86 °F (30 °C) and 20–30%, respectively. The skinks were exposed to a 12 h photoperiod using ambient fluorescent lighting; no ultraviolet B lighting was provided. Animals were provided ad-libitum chlorinated tap water daily and fed three times a week with 30 g of wet cat food (Wellness, Burlington, MA, USA). The skinks were fasted the day before any sedation trial. The sedation trials were performed in a room with an ambient temperature of 75 °F (23.9 °C), but the skinks were kept in an incubator (TLC-40; Brinsea, Titusville, FL, USA) set at 85 °F (29.4 °C) during trials to maintain a consistent body temperature. The skinks were only removed from the incubator for the initial injection and for measuring the physiologic parameters at baseline and every 5 min until the trial was completed; the skinks were returned to the incubator after each sampling period. 

### 2.2. Pilot Sedation Trials

A set of pilot studies were done to determine the final doses for the primary sedation trials. Three drugs were used in four different combinations for the pilot studies: alfaxalone (Alfaxan, 10 mg/mL; Jurox Inc., Kansas City, MO, USA), dexmedetomidine (Dexmedesed, 0.5 mg/mL; Dechra Veterinary Products, Overland Park, KS, USA), and midazolam (5 mg/mL; Hikma Pharmaceuticals, Berkeley Heights, NJ, USA). Four doses of alfaxalone; two doses of alfaxalone with midazolam; two doses of dexmedetomidine with 1mg/kg midazolam; and one dose of alfaxalone, dexmedetomidine, and midazolam were evaluated (in total, 18 trials, Table 1). These doses were selected based on the authors’ previous clinical experiences with skinks. All injections were given SC using one syringe in the right flank, caudal to the right forearm. Skinks were randomly selected for each pilot study using a random number generator (random.org; accessed on 14 August 2023), and two skinks were used for each dose tested. A 2 week washout was provided between each pilot study. Moreover, all doses were tested in randomized dosing order, and the individual assessing the skink during each trial was blinded to the drug and dose because a second individual obtained and administered the drugs. 

Prior to starting any trial, a baseline HR, RR, palpebral reflex, righting reflex, escape reflex, toe pinch withdrawal reflex, tongue flicking, and the possibility of intubation were recorded. Heart rate was measured using a crystal Doppler (Parks Medical Electronics, Aloha, OR, USA) for 15 s and multiplying the number by 4 to determine beats/minute, while RR was measured by counting rib excursions for one full minute to limit the impact of breath-holding behavior. The palpebral reflex was evoked by touching a cotton-tipped applicator to the medial and lateral canthus of the eye and noting a blink. The righting reflex was evoked by placing the skink on its back and measuring the time required to return its body to a sternal position. The escape reflex was induced by pinching the distal third of the tail and evaluating whether the skink moved away from this stimulus. This reflex was also used to determine the suitability for collecting radiographic images using the sedation protocols. For the toe pinch withdrawal reflex, the toes of the hindlimbs were used. A padded hemostat was used to apply pressure to a digit to determine if the skink withdrew the leg as an indicator of deep pain. The tongue flicking was based on whether the tongue moved freely. For the reflexes and possibility of intubation, a 1–3 ordinal scale was used: (1) reflex was present (<1 s) or unable to intubate at all, (2) reflex was delayed (1–4 s) or unable to intubate easily but might be possible with physical restraint of the mouth, and (3) reflex was absent (>5 s) or able to intubate smoothly. The HR, RR, reflexes, and attempts at intubation were recorded every 5 min until all 5 reflexes returned to baseline (score = 1). Full sedation was determined to be when the righting and escape reflexes were both scored as 3 (absent). All skinks recovered uneventfully from the pilot studies.

### 2.3. Primary Sedation Trials

The doses used in the pilot trials that had the least impact on HR and RR, provided sedation, and consistently allowed for intubation were selected for the final complete sedation trial evaluating all 11 skinks. The final doses selected were as follows: 20 mg/kg alfaxalone [A]; 10 mg/kg alfaxalone and 1 mg/kg midazolam [AM]; 0.1 mg/kg dexmedetomidine and 1 mg/kg midazolam [DM]; and 5 mg/kg alfaxalone, 0.05 mg/kg dexmedetomidine, and 0.5 mg/kg midazolam [ADM]. The injection site and route were the same as in the pilot trials. All 11 skinks were randomly assigned to the order in which they received each combination, and a 2 week washout was provided between trials. Two to four combinations were used on each trial day using three to six individuals, and the individuals assessing the skinks were blinded to the drug and dose. To avoid a potential bias attributed to a refractory response to the sedatives over time, the order of the sedatives used was included as an independent variable in the statistical modeling. The primary sedation trials were performed under the same environmental conditions as described for the pilot trials. Similar to the pilot study, all parameters were measured at baseline and every 5 min until the skink recovered. However, unlike the pilot study, 0.05 mg/kg flumazenil (0.1 mg/mL; Hikma Pharmaceuticals) and 0.5 mg/kg atipamezole (Revertidine, 5 mg/mL; Modern Veterinary Therapeutics, Miami, FL, USA) were given 60 min SC after induction to skinks receiving dexmedetomidine or midazolam, respectively, if the skinks were not recovered. The time to loss of righting reflex, time to intubation, time to recovery, duration of intubation, and duration of the loss of righting reflex were also analyzed. Intubation was assessed only for feasibility and the tube was not maintained. Attempts were made every 5 min. Again, recovery was defined as the time when all reflexes returned to baseline (ordinal score = 1). Blood samples were collected from the ventral tail vein of six randomly selected skinks using a 25-gauge needle fastened to a 1 mL syringe to measure blood gases at baseline, time of full sedation, and recovery for each of the drug combinations. The same 6 skinks were sampled for each drug combination to minimize individual physiologic variation. The venous gas analysis was conducted using the VetScan i-STAT 1 (Abaxis, Union City, CA, USA) and CG4+ cartridge. The body temperature for the i-STAT was corrected to 86 °F (30 °C) to match the skink body temperature. No more than 0.15 mL of whole blood was collected for each sample, and thus, blood sampling was limited to <1% of body weight.

### 2.4. Statistical Analysis

The sample size for the primary trials was based on the following a priori data: an alpha = 0.05, power = 0.8, an expected difference in full sedation time between drug combinations of 8 min, and standard deviations between groups of 8 min. The sample size for the blood gas component of the study was based on an alpha = 0.05, power = 0.8, a mean difference in pH between baseline and full sedation of 0.3, and a standard deviation of 0.15. The minimum sample sizes for each of these comparisons were 10 and 5, respectively. Because our IACUC recommends a 10% buffer to minimize the likelihood of a type II error, an additional subject was added to each group to obtain the final numbers of 11 and 6 for the primary sedation trial and blood gas component, respectively. Sample size calculations were performed using MedCalc^®^ Statistical Software version 22.006 (MedCalc Software Ltd., Ostend, Belgium; https://www.medcalc.org; accessed on 14 August 2023). The distributions of the continuous data and their residuals were assessed for normality using the Shapiro–Wilk test, q-q plots, skewness, kurtosis, and histograms. Data that met the assumptions of normality are reported by the mean ± SD, while non-normally distributed data are reported by the median (interquartile range [IQR]) values. When the residuals met the assumption of normality, linear mixed models were used to determine if there were differences in the outcome data (HR, time to loss of righting reflex, duration of intubation, duration of loss of righting reflex, and blood gases values [pH, PCO_2_, HCO_3_, and lactate]) by group (A, AM, DM, and ADM), time (0–60 min with 5 min intervals), and order drugs (1–4). Skink was entered into the model as the random factor, and group, time, order, and their interaction terms were entered as fixed factors. Body weight was also added to the model as a covariate. The Akaike’s information criterion was used to help assess model fit. Because the residuals for RR were not normal, a generalized linear mixed model was used to determine whether the fixed factors group, time, order, and their interaction terms impacted RR. Skink served as the random variable in the model and body weight was added as a covariate. Bonferroni’s multiple comparisons test was used as a post-hoc test when differences were noted. The time to intubation was analyzed using a generalized linear mixed model with group and order as fixed variables and body weight as a covariate. Dunn’s multiple comparison test was used as a post-hoc test. Ordinal data, including all reflexes and the possibility of intubation, were analyzed using the duration of the loss of each reflex as a continuous outcome variable using generalized linear mixed models. Group and order were included in the model as fixed factors and body weight as a covariate. Data were analyzed using SPSS V27.0 (IBM Statistics, Armonk, NY, USA) and GraphPad Prism V9.0 (GraphPad Software, San Diego, CA, USA). A *p*-value < 0.05 was used to determine statistical significance.

## 3. Results

All skinks recovered well without any adverse effects from both the pilot and primary studies. Following SC injection, erythema was immediately noted in more than half of the trials with all drugs but had dispersed in all cases within 10 min.

### 3.1. Pilot Sedation Trials

Every skink that received DM was sedated for over 180 min, regardless of the doses, and thus were reversed with flumazenil and atipamezole. Within the pilot trials, times to recovery for the two skinks in each trial were as follows: 25 and 30 min for 10 mg/kg alfaxalone; 35 and 35 min for 15 mg/kg alfaxalone; 30 and 50 min for 17 mg/kg alfaxalone; 60 and 130 min for 20 mg/kg alfaxalone; 60 and 65 min for 10 mg/kg alfaxalone with 1 mg/kg midazolam; 60 and 65 min for 15 mg/kg alfaxalone with 1 mg/kg midazolam; 170 and >180 (reversed at 180 min with atipamezole) minutes for 0.05 mg/kg dexmedetomidine with 1 mg/kg midazolam; >220 (reversed at 220 min with atipamezole and flumazenil) and 180 min for 0.1 mg/kg dexmedetomidine with 1 mg/kg midazolam; and 100 and 105 min for 5 mg/kg alfaxalone, 0.05 mg/kg dexmedetomidine, and 0.5 mg/kg midazolam. All lizards receiving DM or ADM were bradycardic and bradypneic at the time all their reflexes returned to baseline, unless they were reversed. The loss of righting reflex was recorded in all skinks in the pilot study except one of the skinks receiving 10 mg/kg alfaxalone. Intubation was only possible in 100% (2/2) of the pilot animals for 20 mg/kg alfaxalone, 10 mg/kg alfaxalone and 1 mg/kg midazolam, 0.1 mg/kg dexmedetomidine with 1 mg/kg midazolam, and 5 mg/kg alfaxalone with 0.05 mg/kg dexmedetomidine and 0.5 mg/kg midazolam. Intubation was not always possible with the other drug combinations (10 mg/kg alfaxalone [0/2]; 15 mg/kg alfaxalone [1/2]; 17 mg/kg alfaxalone [1/2]; and 0.05 mg/kg dexmedetomidine with 1 mg/kg midazolam [1/2]).

### 3.2. Primary Sedation Trials

In the primary trials, differences in the reflexes were found between drugs and over time (Figure 1, Table 2). The times to the loss of the righting reflex were the shortest in A and AM, although this difference was not significantly different (*p* = 0.06). All skinks lost their righting reflex during the trials except a single skink in the DM group; this animal was excluded from the final calculation because the skink never achieved that outcome. There was a significant difference in the duration of the loss of the righting reflex between drugs (*F* = 5.22, *p* = 0.011), with ADM being significantly longer than A (*p* = 0.01) and DM (*p* = 0.046). The total loss of the escape reflex in all skinks was only observed in the ADM skinks, whereas in the other drug groups, it occurred in most of the skinks (A: 8/11; AM: 9/11; DM: 8/11). The duration of the loss of the escape reflex was different by drug (*F* = 8.6, *p* = 0.001). The duration of the lost escape reflex was the longest in ADM (55 [45,46,47,48,49,50,51,52,53,54,55,56,57,58,59,60] min) and was almost equal to the entire sedation period after induction. The complete loss of the reflex in all 11 skinks was only obtained in ADM. Post-hoc tests confirmed significant differences in the escape reflex between A-ADM (*p* = 0.001) and AM-ADM (*p* = 0.001).

The palpebral reflex was never lost in the skinks receiving A (0 [0–0] min) or AM (0 [0–0] min) but was lost for 50 (40–55) min and 55 (50–55) min in all skinks receiving DM and ADM, respectively. There was a significant difference in the duration of the loss by drug group (*F* = 102.5, *p* < 0.001), with pairwise differences found between all groups (all *p* < 0.001) except A-AM and DM-ADM (*p* > 0.99). The toe pinch withdrawal reflex was never lost for skinks given A (0 [0–0] min) or AM (0 [0–0] min) but was lost for 10 (0–40) min and 40 (15–50) min for DM (8/11; 73%) and ADM (9/11; 82%), respectively. Again, a significant difference in the duration of the loss by drug group was observed (*F* = 14.5, *p* < 0.001), with pairwise differences (all *p* < 0.05) between all groups except A-AM and DM-ADM (all *p* > 0.99). The duration of the loss of the tongue flicking was significantly different by drug (*F* = 14.0, *p* < 0.001). The durations of time for the lost tongue flicking for each drug were as follows: A, 25 (0–30) min; AM, 30 (15–35) min, DM, 50 (25–60) min, and ADM, 55 (50–60) min. Significant differences were noted between all groups (all *p* < 0.05), except A-AM (*p* = 0.196) and DM-ADM (*p* = 0.09).

There was a significant difference in time to intubation by drug (*F* = 16.9, *p* < 0.001), with skinks receiving DM taking longer than A, AM, and ADM (all *p* < 0.001). The skinks that could not be intubated (>60 min) were again excluded from the analysis. The duration of intubation was significantly different by drug group (*F* = 11.9, *p* < 0.001), with differences noted between ADM and A (*p* < 0.001), AM (*p* = 0.002), and DM (*p* < 0.001). One skink in the A group and two in the DM could only be intubated for 5 min, and the time that was available for the two DM skinks was 45 and 50 min. Since two skinks in the DM group could not be intubated, 4/11 DM skinks only provided 0–5 min of time for intubation. The remaining 7/11 DM skinks could be intubated for more than 40 min. The duration of intubation was consecutive in A, AM, and ADM, whereas consecutive intubation was only achieved in 3/11 DM skinks. All ADM skinks provided more than 40 min of intubation, except for a single animal where only 20 min was possible.

The majority of skinks administered A and AM recovered prior to the 60 min deadline, except for a single skink given A (130 min) and two skinks given AM (reversed at 60 min). None of the skinks receiving DM or ADM recovered by the 60 min deadline, and thus, all 11 animals in both treatments received the reversals for dexmedetomidine and midazolam. All skinks recovered within 10 min of receiving the reversal agents.

Heart rate and RR significantly decreased after injection in all groups (Figure 2; Appendix A). The distribution of HR was different based on drug (*F* = 654.5, *p* < 0.001), time (*F* = 42.68, *p* < 0.001), and drug × time (*F* = 5.4, *p* < 0.001). The RR was also significantly different based on drug (*F* = 129.0, *p* < 0.001), time (*F* = 37.6, *p* < 0.001), and drug × time (*F* = 2.1, *p* < 0.001). The order the drugs were provided over the course of the trial did not affect HR or RR (all *p* > 0.05). Mean ± SD and median (IQR) baseline HR and RR were 79.7 ± 6.2 and 28.5 (20–39.8), respectively. In A and AM skinks, the lowest median HR was 56 for both groups, whereas the lowest HR in the DM and ADM groups were 24 and 28, respectively. Compared to the baseline HR values, a significant difference was observed in A for 15–45 min, AM for 15–40 min, and both DM and ADM for 10–60 min (all *p* < 0.05). The RR was also confirmed to be significantly different from baseline in A for 10–55 min (all *p* < 0.05), AM for 10–40 min (all *p* < 0.05), and DM and ADM for 5–60 min (all *p* < 0.001). The lowest median RR for the A and AM groups were 9 and 7, respectively, while the lowest RR in the DM and ADM groups was 0.

There were significant differences in pH, PCO_2_, and lactate over time (all *p* < 0.001; Figure 3; Appendix A). For pH and PCO_2_, values at the time of full sedation were significantly lower and higher, respectively, from baseline and recovery values (all *p* < 0.001); lactate was only found to be significantly higher at full sedation compared to baseline (*p* < 0.001). The lactate was increased at recovery but was not statistically significant from baseline (*p* = 0.054). A significant difference for the interaction drug × time was only found for PCO_2_ (*F* = 3.2, *p* = 0.008). When analyzing by each drug, DM showed differences between baseline and full sedation (*p* = 0.023), and ADM at full sedation demonstrated a significantly higher value from baseline and recovery (all *p* < 0.001).

## 4. Discussion

The results of this study partially proved our first hypothesis that all four drug protocols would be safe and provide sufficient sedation to allow for minor procedures such as intubation, taking radiographs, and blood collection. Sedation aims to achieve central nerve system (CNS) depression, resulting in an awake but calm state in which the animal is generally uninterested in its surroundings but responsive to noxious stimulation [113]. Even though every regimen induced sedation in this species, there was a noticeable difference between these combinations, including the depth of sedation, duration of time the skinks could be intubated, and duration of sedation provided, with some cohorts not fully being sedated. From the authors’ experience, it is not uncommon to observe within species variation in reptiles when using the same protocol. While we controlled temperature to minimize that impact on metabolism, a recent study implicated obesity and right-to-left cardiac shunts as reasons induction times for anesthesia can be prolonged in reptiles [114]. It is possible that differences in body fat between animals, shunting, and other physiologic differences led to some of the variability found in this study.

In the pilot study, a dose-dependent effect of alfaxalone was confirmed for the duration of sedation, which allowed us to select a dose that would provide the most complete sedation as a single-agent trial. The DM and ADM doses selected for the primary trial provided more than 100 min of sedation in the pilot trials; thus, they could prove valuable for longer minor procedures, if desired. The doses of A and AM selected for the primary trial offered appropriate time for minor procedures meant in this study, especially if reversals are not available. If AM is utilized over A alone, a 50% lower alfaxalone could be used, and the midazolam can be reversed if desired. Although the durations of sedation were adequate at less than 60 min for the desired procedures, considering that there are no reversal agents available for alfaxalone itself, DM or ADM may be chosen to provide that option. Additionally, in the two groups given dexmedetomidine, all skinks had decreased HR and RR even when reflexes returned to baseline values, so reversal is recommended when the goal for sedation is achieved. A study conducted in Argentine tegus found that a combination of 0.2 mg/kg dexmedetomidine and 1 mg/kg of midazolam provided sedation for 80–350 min [70]. The dose of dexmedetomidine used in the tegus was double the dose used in our study, and both the HR and RR were significantly decreased for 6 and 4 h post-sedation, respectively, even though the righting and escape reflexes were returning after 100 min. In our pilot trials, all skinks receiving DM and ADM regained those reflexes at around 80–90 min as well.

The second hypothesis was also partially proven because the reflexes did decrease within 5–10 min of SC injection in 75% of skinks, 81.8% when excluding the DM group. The time to loss of righting reflex and intubation was faster and smoother in the groups that received alfaxalone (A, AM, and ADM) compared to the group that did not (DM). This rapid onset of SC alfaxalone has also been noted in ball pythons (*Python regius*) and leopard geckos (*Eublepharis macularius*) [18,35]. As our findings have proved, these SC drugs can induce effective sedation in blue-tongue skinks and avoid unnecessary pain caused by the large volume being injected IM. The time to intubation in A, AM, and ADM was reasonable, and our ability to perform the intubation was considered straightforward. However, in DM-treated skinks, the time to intubation was highly variable. Among the four DM individuals that could not be intubated or experienced delayed intubation, one of the skinks also did not lose the righting reflex, and the escape reflex was retained in two individuals. These findings affirm that the ability to intubate does not necessarily agree with the loss of righting and escape reflexes. Despite successful initial intubation or the loss of other reflexes, skinks given DM still occasionally reacted to touching their mouths and closed them. This was why consecutive intubation was only achieved in three DM skinks during the primary trials. For intubation, a sensory blockade around the mouth and tongue was required; however, in A and AM skinks, a loss of pain response was not necessary for intubation, and in some DM individuals, intubation was not possible even when the tongue could not move. These results suggest that analgesia is not a prerequisite for intubation.

All or most A and AM skinks maintained their palpebral and toe pinch withdrawal reflexes, contrary to those of the DM and ADM groups. Similarly to our study, 9 mg/kg IV alfaxalone in blue-tongue skinks did not affect the animals’ responses to interdigital pinching [3]. Based on these results, alfaxalone should not be used as a sole agent for sedation protocols where a painful response may be elicited. In the present study, the addition of dexmedetomidine appeared to provide a certain level of analgesia for 10 (0–40) min in the DM and 40 (15–50) min in the ADM groups. The analgesic effect of α_2_-adrenoceptor agonists is provided through supraspinal and spinal mechanisms [115]. The α_1_-adrenoceptor agonist can also provide analgesic effects like an α_2_ agonist, but only at high doses, making them clinically restrictive [116]. Although a direct comparison has not been made with other α_2_-adrenoceptor agonists, dexmedetomidine (medetomidine) has a much greater (10 times more than xylazine) α_2_ selectivity, so it is preferred when pain control is required [117,118]. Unfortunately, the use of α_2_ agonists as sole analgesics is not recommended because of their negative cardiovascular effects [115]. Therefore, they are commonly used in combination with ketamine or opioid analgesics to produce synergistic effects that can be achieved while reducing the doses of each drug [115]. Dexmedetomidine has also been shown to have an opioid-level analgesic effect in dogs when used in combination with ketamine [119]. Regardless, additional analgesics are recommended for blue-tongue skinks because some of the animals receiving dexmedetomidine reacted to painful stimuli. A limitation of these trials was that we only tested a deep pain response by pinching the toe; however, other forms of superficial, thermal, or electrical stimuli can also induce pain. The antinociceptive effects of a DM or ADM combination evoked by thermal, superficial, or deep pain stimuli have been observed in ball pythons, Argentine tegus, and American alligators (*Alligator mississippiensis*) but not in leopard geckos [18,35,70,81]. Only one-third of leopard geckos that received DM lost their superficial skin pinch response and none of the lizards lost their deep pain response. Future studies with blue-tongue skinks should be done to further evaluate and characterize how this species responds to painful stimuli.

Anesthesia is a state of unconsciousness characterized by controlled but reversible CNS depression and perception in which the patient is not awakened by noxious stimuli [113]. According to these criteria, we could achieve some level of anesthesia using ADM but not in A, AM, or DM. Skinks given DM appeared immobile but were somewhat sensitive to stimuli, staying unresponsive when carefully turned over but responding to exaggerated movements when flipped. They also displayed heightened sensitivity and aggressive defensive responses, especially during recovery. These behavioral changes have also been reported in Argentine tegus when using dexmedetomidine [70]. Such responses were not observed when the skinks received ADM, and it was likely through the synergism with the alfaxalone and the deeper sedation observed when all three drugs were used together that these behaviors were eliminated.

Heart rate and RR were significantly decreased in all drug groups, but the reductions were more prominent in the DM and ADM groups, indicating a profound suppressive effect of dexmedetomidine. While this finding was consistent with our third hypothesis, we did also find that HR and RR decreased in the skinks receiving the drug combinations without dexmedetomidine. A decrease in HR from alfaxalone alone has been documented in green iguanas (*Iguana iguana*) using 20 and 30 mg/kg IM and estuarine crocodiles (*Crocodylus porosus*) using 3 mg/kg IV, but conversely, no change in HR was reported in corn snakes (*Pantherophis guttatus*) given 5, 10, and 15 mg/kg SC and bearded dragons (*Pogona vitticeps*) given 15 mg/kg SC [13,14,40,47]. Since all skinks given A or AM maintained their HR > 50 beats/minute, the level of suppression was not considered a serious abnormality. The possibility that the baseline HR and RR values were falsely elevated because of manual handling and excitement should also be considered. It has been reported that even gentle handling increases the HR of the green iguanas [120]. A previous study evaluating a low IV dose of alfaxalone (9 mg/kg) in blue-tongued skinks found that it did not cause noticeable respiratory depression [3]. However, alfaxalone has been associated with decreasing the respiratory rate in other reptiles, including green iguanas, ball pythons, loggerhead sea turtles (*Caretta caretta*), corn snakes, and bearded dragons [8,13,22,25,40,47]. Compared to α_2_ agonists, most of them given 20 mg/kg or less of A maintained respiration rates >6 breaths/minute, similar to the skinks in the present study. However, apnea or bradypnea has been observed in green iguanas and ball pythons given very high doses of A (30 mg/kg) [13,25]. According to one study conducted on bearded dragons comparing injection routes, IV alfaxalone caused the biggest respiratory suppression over other routes [47]. The species differences noted in these studies further reinforce the importance of closely monitoring the HR and RR of reptiles undergoing sedation or anesthesia protocols because we cannot expect all animals to respond similarly to these drugs.

Using dexmedetomidine (DM and ADM), HR decreased to <30 beats/minute and the RR plunged to <2 breaths/minute, representing a >60% decrease in HR and a >90% decrease in RR from the baseline values. Moreover, the period showing a significant difference in HR and RR compared to the baseline values in the DM and ADM groups was twice that of A and AM. This influence on the HR and RR by an α_2_ agonist has been previously reported in reptiles, including ball pythons, leopard geckos, Argentine tegus, and red-footed tortoises (*Geochelone carbonaria*) [18,70,73,80]. These cardiovascular effects were found to be minimized in dogs by giving these drugs via IM or SC routes rather than IV, and at lower doses [121]. However, a significant decrease in HR and RR was still observed in the skinks even when injected SC. This could be related to dosing, as the doses commonly used in reptiles are 10 times higher than that recommended for dogs. Similarly, in humans, dexmedetomidine was found to be the least respiration-depressing anesthetic among those compared; however, the dose used in humans was 100 times lower (1 µg/kg) than that commonly used in reptiles [122]. While these skinks did experience significant reductions in their HR and RR, they did all have unremarkable recoveries and continue to be healthy more than 8 months after the completion of the study. A few reports have described successful sedation in reptiles using dexmedetomidine without alterations in the RR, including leopard geckos and yellow-bellied sliders (*Trachemys scripta scripta*) [18,62]. In those studies, the doses used for DM were similar to what we used in the skinks, while the dexmedetomidine, at twice the dose used in skinks, was combined with ketamine in the slider turtles. Again, these findings further reinforce the importance of developing species-specific sedation protocols to identify potential complications that might occur so that the clinical team managing the reptile patient can provide appropriate corrective measures (e.g., reversal) if needed.

Although the body temperature of the skinks was not measured due to the influence of ambient temperature on reptiles, it is known that sedation with DM in dogs results in a decrease in body temperature [123]. Additionally, since drug metabolism in reptiles can be significantly impacted by body temperature, it is important to maintain the reptiles at a similar environmental and, thus, body temperature in experimental studies [12]. A study of red-eared sliders found that the administration of alfaxalone at 20 °C resulted in significantly longer recovery times and lower heart rates compared to 30 °C. Even with the same dose of alfaxalone, none of the sliders could be intubated at the higher temperature with a dose of 10 mg/kg, and only 30% could be intubated with 20 mg/kg. However, at the lower temperature, 80% could be intubated with the 10 mg/kg dose and 100% with the 20 mg/kg dose. In the present study, the skinks were maintained at the same temperature and exposed to all four sedation protocols to reduce within-animal and temperature-based biases. However, if the DM or ADM protocols reduced body temperatures to lower than the environmental temperature, it could have impacted the skinks. In the future, measuring the core body temperatures of animals being exposed to these sedation protocols could be used to determine whether body temperature is impacted so that corrective measures, such as increasing the environmental temperature, can be implemented.

Given that the authors could only sample a subset of the skink for blood gases, two studies conducted on eastern copperheads (*Agkistrodon contortrix*) and eastern ratsnakes (*Pantherophis alleghaniensis*) were referenced for comparison [124]. The blue-tongued skinks in the present study had slightly higher pH values at baseline (average 7.41) compared to the eastern copperheads but similar values to the eastern ratsnakes. The lowest pH recorded in a skink at full sedation was 7.02 with a PCO_2_ of 50.7 and HCO_3_ of 14.8. When the skink recovered, the pH returned to the baseline range, indicating respiratory acidosis due to reduced RR. The pH significantly decreased at the point of full sedation and then increased at recovery, while there was a corresponding initial increase in PCO_2_ due to the decreased RR, followed by a decrease in PCO_2_ as the RR increased at recovery. Since blood was collected when full sedation was first recorded (approximately 20 min after injection), the parameters reported may not represent the peak or trough values during sedation. However, the marked decrease in pH and increase in PCO_2_ indicated that these sedatives indeed impacted respiratory ventilation. Although the critical limits for these two parameters in reptiles have not been reported, this point should be emphasized in patients where decreased ventilation could be fatal.

The alterations in the blood gases aligned with the significant differences noted over time at the individual drug levels for both DM and ADM, where the most severe respiratory depression occurred. No changes in pH, PCO_2_, or lactate were observed in dogs anesthetized with isoflurane during continuous infusion with dexmedetomidine [125]. However, it is again worth noting that the dose used was 10 times lower than the single dose administered in reptiles. Additionally, the anesthesia in dogs was maintained using isoflurane with tracheal intubation, which provided adequate oxygen even though they did not provide intermittent positive pressure ventilation (IPPV). In our study, a single injection at a lower concentration did not achieve sufficient sedation, indicating the need for further research on the effects of continuous low-dose infusion on respiration and blood gases. A decrease in blood pH has also been described in loggerhead sea turtles given 10 mg/kg alfaxalone IV, and the decreased pH was attributed to a marked decrease in RR and thus increase in PCO_2_ [22]. The turtles were given IPPV, but it was not clear whether the resolution of hypoxemia was due to manual ventilation or to the return of spontaneous breathing. One study conducted in rattlesnakes anesthetized with propofol demonstrated that IPPV significantly affected blood gas parameters [126]. Snakes that received fewer mechanical ventilation compared to baseline had significantly lower pH and PaO_2_ and higher PaCO_2_ than subjects that received more ventilation. Based on these findings, it is recommended to compensate for the physiological changes identified in our study through appropriate IPPV when using these sedation protocols.

Lactate is produced as part of glycolysis in an anaerobic environment and is cleared through the liver and kidneys [127]. It is known that excessive excitement, such as escaping from a predator, can rapidly increase oxygen demand and, thus, lactate in reptiles. This has been reported in crocodiles and gopher tortoises, and in both species, individuals that were physically stunned, so that there was no persistent stress, or that were less stressed due to restricted movement by trapping, showed a smaller increase in lactate compared to those in groups that were stressed [128,129]. In our study, the skinks showed significant increases in lactate at full sedation but subsequently decreased by recovery. The increase was most likely attributed to the hypoxia caused by the decreased RR and stress caused by blood collection. The skink with the lowest pH also had the highest lactate concentration. This alteration in lactate likely reflected the reduction in oxygen and the need to buffer the respiratory acidosis. A study in rattlesnakes similarly found that lactate increased and then decreased after anesthesia; however, in that study, there was no significant difference in lactate concentrations across levels of mechanical ventilation, which made analysis of respiration influences difficult [126]. These results further reinforce the value of measuring blood gases in reptile patients to gain a better understanding of the physiologic consequences of sedatives or anesthetics in these animals.

There were several potential limitations with this study. Sample size is traditionally a limitation in these types of clinical trials. However, to increase the power associated with the study, a complete crossover study was done to reduce within-skink variability between sedation protocols. Moreover, since we were able to reject the null hypotheses for the primary hypotheses we tested, the sample size was sufficient to limit the impact of a type II error. For the blood gas analysis, we collected venous blood gas samples. This is commonly done for reptiles because of the challenges in obtaining arterial samples without invasive methods (e.g., surgical cut-down). While arterial blood gases are preferred, venous blood gases are less of a concern for reptiles, especially three-chambered heart species, because of the inherent mixing of blood during systole. In our case, the venous blood gases still provided an opportunity to characterize trends over time and between sedation protocols.

## 5. Conclusions

Overall, blue-tongued skinks were sufficiently sedated to perform an examination, collect blood, and be positioned for radiographs using all four sedation protocols. Intubation was also achieved using all four sedation protocols; however, with DM (dexmedetomidine-midazolam), 4/11 animals could not be intubated or intubated over 45 min. The authors found that ADM (alfaxalone-dexmedetomidine-midazolam) provided the best overall sedation for performing these minor procedures, followed by AM (alfaxalone-midazolam). These findings further reinforce the importance of using multiple drugs in combination at lower doses to achieve a synergistic effect and minimize side effects.

## Figures and Tables

**Figure 1 animals-14-02636-f001:**
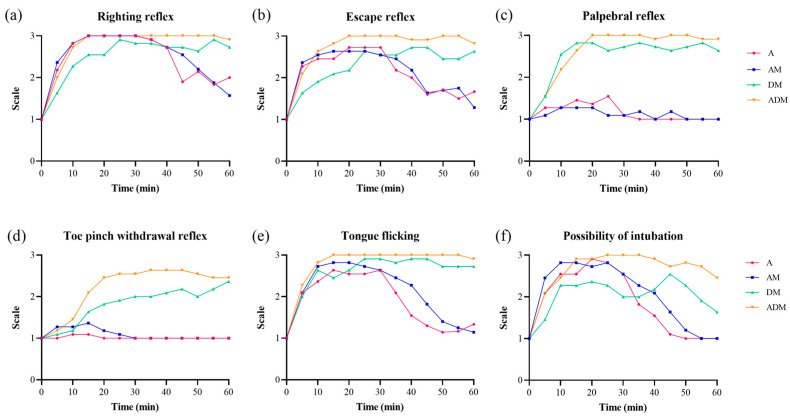
Frequencies for all reflexes by drugs over time for 11 common blue-tongued skinks. (**a**) The loss of righting reflex was observed in all skinks except one skink which was given dexmedetomidine-midazolam (DM). (**b**) The complete loss of the escape reflex in all 11 skinks was only achieved in alfaxalone-dexmedetomidine-midazolam (ADM). Most of the skinks receiving alfaxalone (A) or alfaxalone-midazolam (AM) maintained their (**c**) palpebral reflexes and (**d**) toe pinch withdrawal reflexes. (**e**) The complete loss of tongue flicking was found in DM and ADM, and (**f**) the possibility of intubation was highest in ADM followed by AM and A.

**Figure 2 animals-14-02636-f002:**
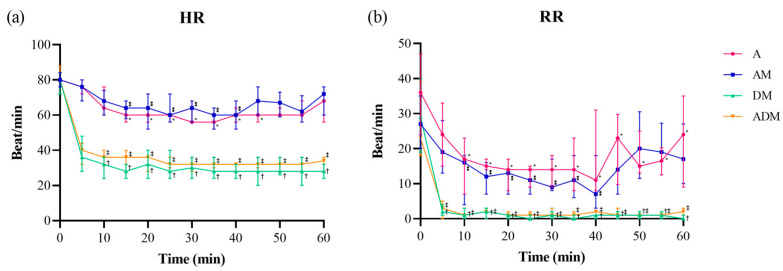
(**a**) Heart rate (HR) and (**b**) respiratory rate (RR) over time for each of the four drugs administered subcutaneously to 11 skinks (A: alfaxalone 20 mg/kg; AM: alfaxalone 10 mg/kg + midazolam 1 mg/kg; DM: dexmedetomidine 0.1 mg/kg + midazolam 1 mg/kg; ADM: alfaxalone 5 mg/kg + dexmedetomidine 0.05 mg/kg + midazolam 0.5 mg/kg). The median (dot) and interquartile range (bar) are shown. Both parameters showed statistical differences by drug, time, and drug × time (all *p* < 0.05). All four drugs showed significant decreases from baseline in HR and RR (*p* < 0.05). However, the duration of the shown differences was almost doubled in DM and ADM. Significantly different time points from the baseline value are marked (* A, ⁑ AM, ^†^ DM, and ^‡^ ADM).

**Figure 3 animals-14-02636-f003:**
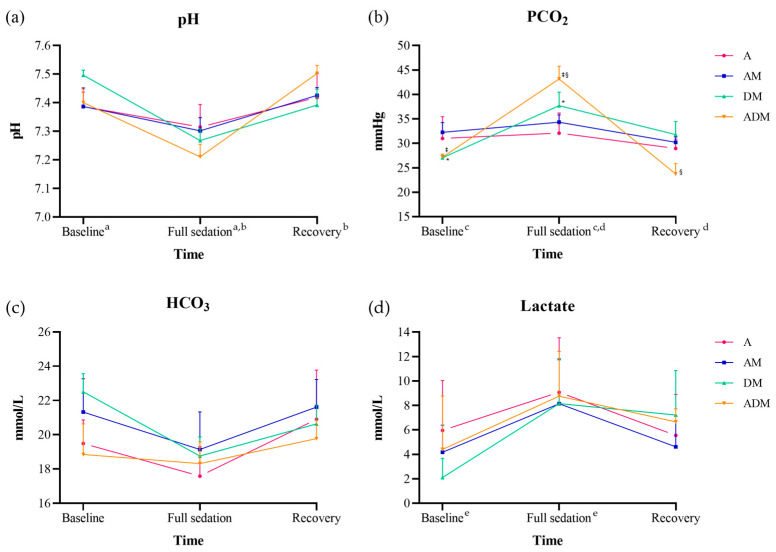
Blood gas results for (**a**) pH, (**b**) PCO_2_, (**c**) HCO_3_, and (**d**) lactate from six skinks administered alfaxalone (A), alfaxalone-midazolam (AM), dexmedetomidine-midazolam (DM), and alfaxalone-dexmedetomidine-midazolam (ADM). The data are reported by the mean (dot) and standard deviation (bar). The pH, PCO_2_, and lactate showed a significant difference by time (all *p* < 0.001), and only PCO_2_ was significant by drug × time (*p* = 0.008). Significant post-hoc comparisons by time are indicated by ^a–e^ (all *p* < 0.001). When analyzing PCO_2_ by each drug, skinks receiving DM significantly increased at full sedation from baseline (*; *p* = 0.023), while skinks receiving ADM showed significant differences between full sedation and baseline and recovery (^‡^, §; all *p* < 0.001).

**Table 1 animals-14-02636-t001:** Description of the drugs, sample sizes, and doses used to determine sedation protocols for common blue-tongued skinks.

Drugs	Pilot	Primary	Alfaxalone	Dexmedetomidine	Midazolam
	N of Skinks				
A					
	2		10 mg/kg		
	2		15 mg/kg		
	2		17 mg/kg		
	2	11	20 mg/kg		
AM					
	2	11	10 mg/kg		1 mg/kg
	2		15 mg/kg		1 mg/kg
DM					
	2			0.05 mg/kg	1 mg/kg
	2	11		0.1 mg/kg	1 mg/kg
ADM					
	2	11	5 mg/kg	0.05 mg/kg	0.5 mg/kg

**Table 2 animals-14-02636-t002:** Descriptive statistics for the durations of time (minutes) that reflexes were lost in blue-tongued skinks following sedation with alfaxalone (A), alfaxalone-midazolam (AM), dexmedetomidine-midazolam (DM), and alfaxalone-dexmedetomidine-midazolam (ADM). The mean ± standard deviation or median (interquartile range) are reported.

Variables	A	AM	DM	ADM
Time to loss of righting reflex	8.18 ± 3.37	8.18 ± 4.05	14.09 ± 6.64	9.09 ± 4.37
Duration of loss of righting reflex *	38.18 ± 8.86 ^a^	41.82 ± 9.83	40.91 ± 18.07 ^b^	55.45 ± 4.98 ^ab^
Duration of loss of escape reflex *	20 (0–35) ^c^	30 (15–35) ^d^	35 (0–45)	55 (45–60) ^cd^
Time to intubation *	10 (5–20) ^e^	5 (5–10) ^f^	10 (10–30) ^efg^	5 (5–15) ^g^
Duration of intubation *	23.18 ± 9.11 ^h^	31.36 ± 10.46 ^i^	28.18 ± 21.24 ^j^	48.18 ± 10.06 ^hij^
Time to recovery ^†^	55 (45–60)	60 (50–60)	60	60

The significant difference (*p* < 0.05) confirmed between groups (*). As DM and ADM groups were reversed at 60 min, time to recovery was not analyzed by group (^†^). ^a^ (*p* = 0.01), ^b^ (*p* = 0.046), ^c,d^ (*p* = 0.001), ^e,f,g^ (*p* < 0.001), ^h^ (*p* < 0.001), ^i^ (*p* = 0.002), ^j^ (*p* < 0.01).

## Data Availability

The original datasets presented in the study are included in the article/Appendix A, further inquiries can be directed to the corresponding author.

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
