# Peer review of "Evaluating the Physiologic Effects of Alfaxalone, Dexmedetomidine, and Midazolam Combinations in Common Blue-Tongued Skinks (Tiliqua scincoides)"

_animals, 2024, doi:10.3390/ani14182636_

Round 1

Reviewer 1 Report

Comments and Suggestions for Authors

Dear authors, please address the following:

Line 117: how did you decide on using 11 individuals, not less, not more, please explain!

Line 128: please provide temperature value also in Celsius

Line 170-171: please mention administration route for these substances

Line 537: approximately  20 minutes after substance administration

Line 595-597: please provide full names of drugs in brackets for DM, ADM and AM

I believe your research is interesting and provides more insight into reptile anesthetic protocols.

Author Response

Reviewer 1:

Line 117: how did you decide on using 11 individuals, not less, not more, please explain!

            A: We specifically addressed why we used 11 animals by providing an a priori sample size estimation described in lines 208-213.

“The sample size for the primary trials was based on the following a priori data: an alpha=0.05, power =0.8, an expected difference in mean sedation between drug combinations of 8 minutes, and standard deviations between groups of 7 minutes. The sample size for the blood gas component of the study was based on an alpha=0.05, power=0.8, a mean difference in pH between baseline and full sedation of 0.3, and a standard deviation of 0.15.” We have also added the following based on reviewer 3’s suggestion: “The minimum sample sizes for each of these comparisons were 10 and 5, respectively. Because our IACUC recommends a 10% buffer to minimize the likelihood of a type II error, an additional subject was added to each group to obtain the final numbers of 11 and 6 for the primary sedation trial and blood gas component, respectively. Sample size calculations were performed using MedCalc® Statistical Software version 22.006 (MedCalc Software Ltd, Ostend, Belgium; https://www. medcalc.org; 2023).”

Line 128: please provide temperature value also in Celsius.

            A: Sorry, we missed that. Thank you for pointing it out. We have added 29.4oC.

Line 170-171: please mention administration route for these substances.

            A: Thank you. We added “SC” to line 192.

Line 537: approximately 20 minutes after substance administration

            A: Is it correct that you recommend adding ‘after injection’ in the bracket? We added it.

Line 595-597: please provide full names of drugs in brackets for DM, ADM and AM

            A: We have added them but do not think this is the correct use of the method of using abbreviations. We ask the editor to give a final determination. It is not typical format to spell out an abbreviation later in a manuscript; however, we are fine if that is what is wanted.

Reviewer 2 Report

Comments and Suggestions for Authors

The study is interesting but needs further clarification and some corrections before it can be considered for publication.

I read with interest the paper entitled “Evaluating the physiologic effects of alfaxalone, dexmedetomidine, and midazolam combinations in common blue-tongued skinks (Tiliqua scincoides)The study is interesting and the data provided can enrich our knowledge of species that are still little studied.

However, the study needs further clarification and some corrections before it can be considered for publication.

The introduction must be small, I do not think it is necessary to list the work done before.

Among the effects of dexmedetomidine, it should be noted that it causes analgesia, an effect we do not have with the other two drugs.

Materials and Methods

Line 117:  Where did the animals in the study come from? Were they used to being handled?

Line 138: The subdivision of groups and methods of drug administration can be confusing to the reader, and it would be good to make this part more schematic.

Line 146: Was the person administering the drugs not aware of the drugs used? But were there differences in the doses (ml) of the drugs?

Line 168: Did you know in advance how long the effect would last? Rewrite the sentence correctly.

Line 185: See previous comment.

Results

Line 232: I find this part “Pilot sedation trials”  confusing and unclear.

Discussion

Line 368: I would delete this sentence, as the diversity of the effect can be found in all species.

Line 384: I do not really agree with this statement because it is true that some drugs can be antagonised but the effect of the other drug will still be present. The duration of the effect of the different drugs should also be evaluated.Also, it has not been considered that not all drugs give analgesia.

Line 512: This reference can be added because environmental temperature is also very important for wild species. doi: 10.7589/2016-06-131. Epub 2017 Jan 25. PMID: 28122191.

Author Response

Reviewer 2:

The introduction must be small, I do not think it is necessary to list the work done before.

            A: We respectfully disagree. An introduction is meant to guide the reader as to the intent of the manuscript. The corresponding author teaches a course on this (18 years) and recommends a minimum 3 paragraph Introduction to introduce the topic, guide the reader to the importance of the research, and outline the objectives/hypotheses. I also have 273 peer reviewed articles, so I have used this method successfully for some time. The introduction in this case introduces the species of lizard (“the who”), the dearth of evidence on sedation for them and the background on the three selected drugs (“the why it’s needed”), and the objectives and hypotheses (“the plan”). Without a specific recommendation on what to reduce, it is challenging to do so. Moreover, for those with little experience in the field, this Introduction should set them up well to understand the research and why we selected it. Since the other reviewers didn’t comment on introduction length, we prefer to leave it unless there are specific examples to consider. Thank you.  

Among the effects of dexmedetomidine, it should be noted that it causes analgesia, an effect we do not have with the other two drugs.

            A: Thank you. Yes, we agree. We had noted this in the original draft as “visceral analgesia” (now line 78).

Materials and Methods

Line 117:  Where did the animals in the study come from? Were they used to being handled?

            A: We meant to add that. Thank you for pointing that out. We have added: “The skinks were obtained from a private breeder and were used to routine handling.”

Line 138: The subdivision of groups and methods of drug administration can be confusing to the reader, and it would be good to make this part more schematic.

            A: Thank you. We originally had it in a draft but removed it. We agree and re-added it into the paper as Table 1. (lines 142-149, Table 1).

Line 146: Was the person administering the drugs not aware of the drugs used? But were there differences in the doses (ml) of the drugs?

            A: Yes, the individual observing the animal did not know what the drugs were because a second individual administered the drugs. Thank you for that suggestion, we have added the following for completeness: lines147-149. Moreover, all doses were tested in randomized dosing order and the individual assessing the skink during each trial was blinded to the drug and dose because a second individual obtained and administered the drugs.

Line 168: Did you know in advance how long the effect would last? Rewrite the sentence correctly.

            A: Thank you. No, we moved the sentence to result section in lines 246-247. Should have been in results. Thanks for the suggestion.  

Line 185: See previous comment.

            A: This is not related to any results. We meant that we used reversals at 60 minutes if the skinks were not recovered, which was different from the pilot study (planned not to use reversals).

Results

Line 232: I find this part “Pilot sedation trials” confusing and unclear.

            A: We are sorry about any confusion but included the pilot data to guide the reader as to how we determined the final doses. Not including the pilot data might lead some to ask, “how did you decide on the doses?”, but then we would be back to including the same results (as we have). Since there were only 2 skinks per group, we thought it best to list the results rather than a table because there is no value in measuring central tendency or dispersion of 2 data points.

Discussion

Line 368: I would delete this sentence, as the diversity of the effect can be found in all species.

            A: Thank you. Deleted.

Line 384: I do not really agree with this statement because it is true that some drugs can be antagonized but the effect of the other drug will still be present. The duration of the effect of the different drugs should also be evaluated. Also, it has not been considered that not all drugs give analgesia.

            A: We are not sure we understand your concerns. Yes, our focus was on the reversal of the drug intended to be reversed (ie, flumazenil for midazolam) but that it would not have an effect on the other drugs (ie, alfaxalone). We did not infer that. We simply said that by using AM instead of A alone you could reduce the dose of A and reverse the dose for midazolam. This is standard practice and why it is preferred (as we referenced in paper) to use synergistic combinations of sedatives versus single drugs. We do appreciate your point about time between A and AM based on total recovery time, so we removed “making shorter sedation available” and replaced it with “if desired” hoping that addresses one of your concerns. Thank you.

Line 512: This reference can be added because environmental temperature is also very important for wild species. doi: 10.7589/2016-06-131. Epub 2017 Jan 25. PMID: 28122191.

            A: Thank you for the recommendation. The reference we included [123] specifically addresses this and we included it because the was conducted in dogs using dexmedetomidine and midazolam, the same combination in our study. Also, as the ambient temperature is especially important in reptiles, we included one more study conducted in red-eared sliders [12] showing different environmental temperatures significantly altered the effect of the same dose of the sedative.

Reviewer 3 Report

Comments and Suggestions for Authors

Dear Authors,

Congratulations on the drafting of this manuscript and for the clarity of the work. I am providing some comments below.

Line 14. Replace “Drugs” with “combinations”.

Line 18. Replace “These” with “this”.

Line 27-28. The sentence is a bit convoluted, as "were most prominent" currently refers to the subject "heart and respiratory rates." I believe the reference is actually to the decrease in rates. Please rewrite the sentence to be grammatically correct. The same applies to lines 462-463.

Line 51 and throughout the text. Move the period after the square bracket.

Line 55 and throughout the text. Move the comma after the square bracket.

Line 63. Alfaxalone is more commonly defined as an anesthetic, not as a sedative, although it can also produce sedation at sub-anesthetic doses.

Line 122. Why was a relative humidity of 20-30% chosen and not a higher level?

Line 124. I would write "ad libitum" in full.

Line 128. Since the temperature was previously reported in degrees Celsius, I would write it here as well. This makes it more accessible to European readers.

Paragraph 2.3. Were 2 weeks also waited for the wash-out period in this case? Were all 4 cocktails administered to all 11 animals? In what order? This is not clear. Please add this information to the text in this paragraph as well.

Line 187. Regarding the “duration of intubation,” what criteria were used to decide when to extubate? Or do you mean that an attempt was performed, and the tube removed each time?

Lines 199-204. The sample size calculation is mentioned, but the number obtained from the analysis is not reported. Additionally, it is not described which software was used for this part of the analysis.

Line 200. What is meant by “difference in mean sedation”? Does this refer to the duration of sedation or the time at which a sedated state is reached? Please clarify. Also, on what basis were these values established?

Line 212. Delete “were used,” as it was already mentioned earlier.

Line 238 and following. Atipamazole -> atipamezole

Lines 417-420. This sentence is of little use to the discussion of this study; I would eliminate it.

Line 532. Cooperheads -> copperheads

Line 533. A pH of 7.02 does not indicate respiratory acidosis, but simply acidemia. That the cause may potentially be respiratory is possible, but the value needs to be associated with pCO2 and HCO3 to speak of respiratory acidosis. If necessary, report these data here in parentheses.

Line 359. The Discussion section is very broad and comprehensive; perhaps, overly so. Some concepts are presented in a very verbose and prolix manner and are repeated in several sections. This makes the text less fluid, and it becomes difficult to perceive which points deserve more emphasis. I would suggest trying to reduce this part by summarizing.

Author Response

Reviewer 3:

Line 14. Replace “Drugs” with “combinations”.

            A: Thank you. Done.

Line 18. Replace “These” with “this”.

            A: Thank you. Done.

Line 27-28. The sentence is a bit convoluted, as "were most prominent" currently refers to the subject "heart and respiratory rates." I believe the reference is actually to the decrease in rates. Please rewrite the sentence to be grammatically correct. The same applies to lines 462-463.

            A: Thank you. Done.

Line 51 and throughout the text. Move the period after the square bracket.

Line 55 and throughout the text. Move the comma after the square bracket.

            A: Thank you. We edited these throughout the text.

Line 63. Alfaxalone is more commonly defined as an anesthetic, not as a sedative, although it can also produce sedation at sub-anesthetic doses.

            A: Thank you. We originally used sedative, as it can be classified, because we didn’t measure true pain and thus anesthesia. However, we revised this to your request.

Line 122. Why was a relative humidity of 20-30% chosen and not a higher level?

            A: That is the ambient room humidity. While not preferred, it is what we have at our facility. While not ideal, no animals have had any issues with dehydration or dysecdysis for over 2 years in the colony. Moreover, the skinks are always provided with a water bowl and the enclosure humidity may be higher but is not measured cage to cage.

Line 124. I would write "ad libitum" in full.

            A: Thank you. We revised it.

Line 128. Since the temperature was previously reported in degrees Celsius, I would write it here as well. This makes it more accessible to European readers.

            A: Yes, thank you. We have added it.

Paragraph 2.3. Were 2 weeks also waited for the wash-out period in this case? Were all 4 cocktails administered to all 11 animals? In what order? This is not clear. Please add this information to the text in this paragraph as well.

            A: Thank you. We have added the following to clarify this in lines 183-187: The injection site and route were the same as in the pilot trials. All 11 skinks were randomly assigned to the order in which they received each combination, and a 2-week washout was provided between trials. Two to four combinations were used on each trial day using three to six individuals, and the individuals assessing the skinks were blinded to the drug and dose.

Line 187. Regarding the “duration of intubation,” what criteria were used to decide when to extubate? Or do you mean that an attempt was performed, and the tube removed each time?

            A: The latter is correct. We did not maintain the intubation and it was attempted every 5 minutes. We added more description in lines 198-200: Intubation was assessed only for feasibility and the tube was not maintained. Attempts were made every 5 minutes.

Lines 199-204. The sample size calculation is mentioned, but the number obtained from the analysis is not reported. Additionally, it is not described which software was used for this part of the analysis.

            A: Sure thing. We added the following to address this question. Sorry, we left this out originally: lines 216-221: The minimum sample sizes for each of these comparisons were 10 and 5, respectively. Because our IACUC recommends a 10% buffer to minimize the likelihood of a type II error, an additional subject was added to each group to obtain the final numbers of 11 and 6 for the primary sedation trial and blood gas component, respectively. Sample size calculations were performed using MedCalc® Statistical Software version 22.006 (MedCalc Software Ltd, Ostend, Belgium; https://www. medcalc.org; 2023).

Line 200. What is meant by “difference in mean sedation”? Does this refer to the duration of sedation or the time at which a sedated state is reached? Please clarify. Also, on what basis were these values established?

            A: Sorry, we meant “full” sedation, similar to the blood gas analysis. This has been corrected. These times were selected because, with 1 to 2 SD, they represent approximately 15-23 minutes variances between drugs. As this is meant to be a clinical study, these times, based on our personal experiences with these and other lizards for procedures, were appropriate for differing between more successful and less successful sedation times.

Line 212. Delete “were used,” as it was already mentioned earlier.

            A: Sorry, we missed this. Thank you. It has been deleted.

Line 238 and following. Atipamazole -> atipamezole

            A: Thank you for finding the typos. We corrected it.

Lines 417-420. This sentence is of little use to the discussion of this study; I would eliminate it.

            A: Thank you. We agree with your suggestion and have deleted the sentence.

Line 532. Cooperheads -> copperheads

            A: Thanks! Corrected.

Line 533. A pH of 7.02 does not indicate respiratory acidosis, but simply acidemia. That the cause may potentially be respiratory is possible, but the value needs to be associated with pCO2 and HCO3 to speak of respiratory acidosis. If necessary, report these data here in parentheses.

            A: Thank you for pointing this out. We should have included those values in this statement. We added these and revised it in lines 543-546.

Line 359. The Discussion section is very broad and comprehensive; perhaps, overly so. Some concepts are presented in a very verbose and prolix manner and are repeated in several sections. This makes the text less fluid, and it becomes difficult to perceive which points deserve more emphasis. I would suggest trying to reduce this part by summarizing.

            A: We intended for the discussion to be comprehensive because we wanted to show the depth of doing a sedation study. There were quite a few objective continuous measures collected from times to sedation and recovery to heart and respiratory rates and blood gases. We attempted to address each of these findings in detail based on our hypotheses and utilizing references certainly required some explanation, but we feel was necessary. We feel that providing more, especially with an open access journal where page costs are not an issue, just delivers more to the reader. We reviewed this over several drafts to limit the wording and arrived at what we submitted. We would be happy to consider specific suggestions but are not sure what to remove without impacting the final document. Since the other reviewers did not comment on the discussion, we hope the reviewer can see that being comprehensive may have more value to the reader as it is all collected in one paper versus needing to seek it out across multiple texts or not being available at all.

We did remove the following in the hopes this addresses some of your concerns: lines 387-388; 392-395; 441-444;447-448; 453-455 and remain open to specific suggestions.

Round 2

Reviewer 2 Report

Comments and Suggestions for Authors

The manuscript has been revised accordingly and is now ready for publication.